# The Impact of a Gradual Healthier Assortment among Vocational Schools Participating in a School Canteen Programme: Evidence from Sales and Student Survey Data

**DOI:** 10.3390/ijerph17124352

**Published:** 2020-06-17

**Authors:** Ellen van Kleef, Florine Kremer, Hans C. M. van Trijp

**Affiliations:** Marketing and Consumer Behaviour group, Wageningen University, Hollandseweg 1, 6706 KN Wageningen, The Netherlands; florinekremer@hotmail.com (F.K.); hans.vantrijp@wur.nl (H.C.M.v.T.)

**Keywords:** healthy school food environment, nutrition guidelines, availability nudge

## Abstract

Many adolescents have poor eating habits. As a major part of their caloric intake takes place at school, the present study aims to examine the effect of increasing the availability of healthier foods in school canteens on sales, student attitude and self-reported behaviour. A quasi-experimental study was carried out at two vocational schools in the Netherlands over a 10-month period, where the visible share of healthier products was gradually or abruptly increased from 60% to 80%. Outcome measures were sales data for healthier and less healthy foods and drinks in the canteens, as well as surveys. The proportion of healthier products sold increased from 31.1% during the baseline period to 35.9% in the final period. A gradual increase led to higher relative sales of healthier products (40%) than an abrupt change (34.5%). Survey data showed that students’ moderate satisfaction remained insensitive to the changes over time. Overall, results suggest that increasing the availability of healthier products in school canteens leads to small positive changes in sales of products, particularly in the product groups beverages and sandwiches. A gradual introduction may ensure that students slowly get used to assortment changes.

## 1. Introduction

Many adolescents and young adults have poor eating habits, in that their diet contains high amounts of energy-dense foods, saturated fat, total sugars, and a lack of dietary fibre [1]. Moreover, the proportion of overweight adolescents and (young) adults has been rising tenfold in the past four decades [2]. Schools are increasingly seen as an ideal setting for implementing interventions to make young people eat and drink healthier as the majority of children, adolescents and young adults can be reached for longer periods and up to 50% of daily caloric intake takes place at school [3]. Special interest groups increasingly call for action to improve school food policies. Additionally, celebrity chef Jamie Oliver received a lot of media attention for his campaign to put healthy foods on the school menu and ban ‘junk food’ [4].

To reduce the prevalence of overweight and obesity, many countries have developed policies and scientifically substantiated nutrition standards and measures to help schools offer healthy foods, drinks or meals to their students [5,6]. These standards and measures typically focus on limiting the provision of unhealthy foods and increasing the availability of healthier choices such as fruits, vegetables, water, and low-calorie snacks and drinks at schools. A systematic review of 16 isolated school food environment interventions showed that modifying the relative availability of food for sale from school food services is associated with higher purchases or consumption of healthy foods by students [5]. Several studies reviewed had methodological limitations such as the lack of high-quality study designs, short intervention or effect measurement periods, and self-reported consumption or purchase data (e.g., food frequency questionnaire). Often, only one or a few product groups are examined, such as drinks, vegetables and fruit, or snacks (e.g., [7,8]). A recent meta-analysis of 91 school food environment policies interventions (55 in US/Canada, 36 in Europe/New Zealand) showed the efficacy of a range of food environment policies, including direct provision, competitive food/beverage standards, and school meal standards [9]. Another review looked at 25 studies reporting results of implementing behavioural nudges in school canteens to encourage healthier eating. These nudges aimed to make healthier choices more convenient, attractive and acceptable, such as changing the order of serving vegetables or prompting children to take fruit. Results of most studies showed an increase in the selection or consumption of target foods [10]. 

Despite the progress made in getting healthier foods and drinks inside schools, the uptake of healthier products remains a challenge. For students, it is possible to pick unhealthy options from an assortment containing both healthy and unhealthy products. Moreover, there is evidence that adolescents compensate for reduced intake of healthy foods at school by increasing consumption at retail stores and (fast food) restaurants outside of schools [11,12,13]. Although increasing the availability of healthy foods at school is promising in term of effectiveness, it is unknown how large effects are when looking at assortments including both healthy and unhealthy options and whether positive changes are sustainable in the longer term. 

This study aimed to examine the effect of increasing the availability of healthier foods and drinks in school canteens on sales for one school year. We define availability as the visible presence of facings of foods and drinks ready for an immediate choice by consumers. In other words, the assortment stays the same, but the proportion of facings of healthier foods and drinks increased across the entire assortment. In this study, we build on the guidelines of the Dutch Healthy Canteen Programme (developed by Netherlands Nutrition Centre) that define and recommend levels of healthy relative to unhealthy foods on display from 60% to 80%, based on Dutch national nutritional guidelines [14]. These canteens were situated at vocational schools in the Netherlands. In the Netherlands, school canteens at secondary and subsequent education sell lunch, snack foods and drinks to students and employees, instead of, or in addition to, what has been brought from home or bought outside the school. 

We expect that a larger visible proportion of healthier products will lead to significantly more sales of healthier products relative to unhealthy products (hypothesis 1). The reason for this is that if healthier products take a large visual part of the assortment space, this may implicitly give the message to customers that these products are in demand (i.e., popular) and attractive. This corresponds with principles of nudging interventions [15]. Nudges preserve the freedom to choose but give a small push in the right direction by increasing the availability, attractiveness or convenience of healthier options [16]. Increasing the availability, in that a product category gets relatively more shelf space, has been shown to influence sales. For example, research in the 1970s showed that giving additional shelf space to fruits and vegetables in a supermarket increased sales by 44% [17]. A study that manipulated both the pricing and availability of healthier items in vending machines resulted in 10–42% higher sales of healthier items, although the effect of lowering prices could not be separated from the effect of increasing availability [18]. Restricting the availability of unhealthy choices in the assortment of vending machines was more effective in stimulating people to choose healthier products from a vending machine than calorie labelling [19]. 

Our second hypothesis in this study is that a step-by-step increase of the availability of healthier foods and drinks, instead of an abrupt increase, will lead to healthier product sales at school and a better acceptance of the new canteen assortment by students. To our knowledge, there is no evidence of the effect of a more incremental versus a disruptive assortment change. Our expectation is based on studies showing negative responses of students after the implementation of large changes in food offerings at school [20]. When students feel that food choices are imposed on them this may threaten their sense of control, potentially leading to psychological reactance [21]. This, in turn, may result in students obtaining their (unhealthy) foods from other channels than the school canteen [22]. Anecdotal evidence indicates that if the rate of changes in the food assortment is too fast, students will start purchasing fast food outside the school and schools may bring back unhealthy foods in their assortment to prevent monetary losses [23]. Hence, this might be prevented by unobtrusively building up the changes in assortment offerings, allowing customers to get used to it. 

An important question is how students experience the assortment changes in the school canteen. Even though the assortment changes are not communicated to them, a healthier assortment may give students the feeling of being steered towards certain choices or simply be seen as less attractive. It is therefore critical to understand the impact on students’ attitudes towards the canteen in terms of satisfaction, healthiness and assortment perceptions. Student purchases could move away from the school canteen because of negative perceptions and be replaced by food brought from home or purchased elsewhere. Then the overall healthiness of the food intake could decrease despite the healthier assortment. To evaluate student’s satisfaction and purchasing habits, a survey was conducted among students throughout the year to better understand the impact of the intervention on their attitude towards the canteen and self-reported purchase behaviour. 

## 2. Materials and Methods 

### 2.1. Setting and Context 

This study was conducted from September 2016 until July 2017 at two vocational school canteens in the Netherlands (34 weeks in total). Vocational schools provide post-secondary professional training for a wide variety of occupations with a focus on practical hands-on applications of skills. Interventions are increasingly encouraged among people with lower and middle levels of education because they often eat less fruit and are more likely to be overweight, partly believed to be caused by less easy access to healthier foods [24]. 

In the Netherlands, there is no national subsidized school lunch program or system of school meals. Instead, students bring their own packed lunch or purchase meals, snacks and drinks at school or at food outlets nearby school (e.g., supermarket, fast-food chain). Typically, prices at schools’ commercial school canteen catering companies are a bit higher than in supermarkets or fast-food chains.

The largest canteen was in the city of Ede in a building where 2339 students follow their education. A smaller canteen was in the city of Velp, in a building where on average 1124 students follow their education. During holiday weeks (seven in total) the canteens were closed. The school canteens were run by one commercial school canteen organisation. The nearest supermarket was 450 meters from the school in Velp and 1200 meters in Ede (Google Maps). In the last months of the study year, a part of the students was not present in the school building as they were doing their internships.

### 2.2. Study Design and Procedures 

The intervention consisted of assortment changes in two vocational school canteens where the visible share of healthier (‘healthier choice’) products was systematically varied according to experimental design (Table 1). The core of the intervention is that the majority of visible food and drinks assortment at the counter in displays, showcases and vending machines is healthier [25]. To achieve that goal, each food and drink in the canteen was categorised as a ‘healthier choice’ product or an ‘unhealthy choice’ product based on Dutch nutrition guidelines. In these guidelines, products are considered to be a ‘healthier choice’ when they are healthy or reasonably healthy (a middle category in the Dutch Nutrition Guidelines) [14] in their product category. Examples of such products are brown or whole wheat bread, dairy drinks without added sugars, and salads. ‘Unhealthy choice’ products are considered unfavourable in their product categories, such as sugary drinks, white bread, and creamy soup, and are called ‘incidental’. This categorisation was based on nutritional guidelines related to energy density, saturated and trans-fat, added sugar and salt of the Netherlands Nutrition Centre (the governmentally-funded organisation responsible for public nutrition education). For example, snacks are considered a ‘healthier choice’ when the total calories per portion is lower than 110 calories, sodium is less than 400 mg per 100 gram and energy percentage for saturated fat is less than 13. For soft drinks, criteria include total calories which should be less than 30 per 100 millilitres. Their guidelines are based on analyses of the Dutch Health Council [26]. 

This categorization is part of the Healthy Canteen School programme developed by the Dutch Nutrition Centre in 2003. The programme can voluntarily be applied for by schools and is supported by various public health and industry stakeholders. This programme provides food- and behaviourally based recommendations for healthier offerings (e.g., nudging techniques) and their uptake by students at secondary and vocational school canteens [14]. Schools that conform to the guidelines can request a Healthy School Canteen Award. In the 2015 covenant agreement for Healthy Nutrition at Schools in the Netherlands, the signing parties (e.g., catering companies and vending machine suppliers) decided to have a minimum of 60% healthier choice products in the visible assortment [25].

The intervention was implemented in four time periods: an initial baseline phase and three experimental phases (Table 1). In the first baseline period 1 (September 2017 until November 2017), the visible share of ‘healthier choice’ products was at least 60%. After these two months, canteen A increased its visible share of ‘healthier choice’ products to about 80%, while canteen B increased its visible share to about 70% for two months (period 2). From January 16, 2017, all canteens had an 80% ‘healthier choice’ assortment (periods 3 and 4). The exact time of change to another condition was done in agreement with canteen personnel so enough products could be purchased. Important to note is that products can be refilled at any time when their availability is low, and students are free to purchase whatever they want. In correspondence with the guidelines, at least one ‘healthier choice’ product was offered in every product category for the entire study period. Additionally, these choices were placed in prominent places. The product groups present in the canteens were: (1) vegetables, (2) dinner salads, (3) fruit, (4) sandwiches, (5) bread without topping, (6) separate sweet bread toppings, (7) separate savoury bread toppings, (8) dairy products, (9) beverages, (10) sweet snacks, (11) savoury snacks, (12) sauces, and (13) soups. During the entire year, prices remained constant. 

Before the intervention started, a meeting was organized for all canteen employees in which the invention was explained. There were weekly price promotions, both for healthier and unhealthy products. Price promotions were similar in both canteens. 

### 2.3. Manipulation Check: Canteen Scan

The proportion of healthier menu items in each category (including vending machines) was determined monthly by the ‘School Canteen Scan’, an online audit tool developed by the Dutch Nutrition Centre. The scan part of the Healthy Canteen School Programme and offers a momentary assessment of the available number and type of products [27]. When the visible share of ‘healthy products’ on display was too low, changes were made in consultation with canteen employees. Only those products are scored that are available at the moment of assessment. Care was taken that the canteen scan was carried out just before lunch visitors would arrive, to guarantee the maximum number of products on display. Products that got low in availability could be replaced. Warm snacks were prepared on the spot and continuously refilled. 

On average, there were 717 healthier choices and 263 unhealthy choices on display in Ede, and 381 healthier and 139 unhealthy choices in Velp. Of these products, the largest categories were beverages and sweet snacks in Ede (50% and 14% of the total number of products on display, respectively), and beverages and dairy in Velp (28% and 29%, respectively). 

### 2.4. Student Survey

During the entire year, a diverse sample of students attending the vocational schools was asked to fill in a brief questionnaire about their canteen experiences. No reference was made to the intervention that took place at that moment to not interfere with the intervention. Potential participants were approached in various ways. In the first two periods, students were initially approached by e-mail, but due to the low response rate, this was not carried out further. From baseline period 1 on, the majority was approached by research assistants in the canteen, classrooms and corridors and asked whether they were willing to fill in a brief pen-and-paper questionnaire. 

Ethical Approval: Ethical approval for this study was not necessary for the Netherlands at the time of data collection. All procedures performed in studies involving human participants were in accordance with the ethical standards of the institutional and/or national research committee (social-ethical committee of Wageningen University) and with the 1964 Helsinki declaration and its later amendments or comparable ethical standards. No informed consent was obtained as the data consists of personally unidentifiable sales data and anonymous questionnaires among students older than sixteen years of age.

### 2.5. Measures

#### 2.5.1. Cash Register Data

Cash register data form the basis for our key dependent variable, namely: “relative proportion of healthier choice products sold of total sales (%)”, reflecting student and employees’ purchases of all products in the canteen. During the entire period, cash register data was obtained at the transaction level and expressed in terms of total assortment as well as for 13 key product categories (vegetables; salads; fruit; sandwiches; bread, sweet and savoury toppings separately; dairy; beverages; sweet snacks; savoury snacks; sauces and soups). Vending machine sales were not included in the dataset due to unreliable data collection in which it appeared unknown what numbers meant. 

Next to more descriptive outputs per period, such as (i) % of healthier choices at display in the entire canteen (a manipulation check), and (ii) length of the period and average number of products sold per week, additional *transaction-level measures* were derived from the cash register data, namely: (i) number of products sold per transaction, (ii) relative proportion of healthier choice products sold per transaction, and (iii) revenues per unique transaction.

#### 2.5.2. Student Satisfaction 

Customer satisfaction was measured in the surveys conducted at each of the four waves of data collection (including period 1). This was done through three constructs each measured with a single item. *Canteen satisfaction* was measured with the single item ‘I am satisfied with this canteen’. *Perceived opportunity to eat healthily* was measured by the single item ‘In this canteen, you can eat healthily’. *Freedom of choice* was measured by the single item ‘I feel free to choose what I like’. All items were answered on a five-point Likert-type response format ranging from ‘totally disagree’ to ‘totally agree’. Respondents also provided information regarding age, gender and type of education followed.

#### 2.5.3. Customer Patronage

Customer patronage was operationalised in multiple ways. After indicating whether they are a student or employee and at which location they followed a study or worked, in an open-ended question “*presence in building*” was operationalised by asking: ‘How many days a week are you in this building?’ *Outlet choice* was measured by asking respondents to indicate where they purchased their food and drinks. The instruction was ‘Indicate where you get your lunch, drinks and snacks’. Lunch was described to reflect a “typical Dutch lunch structure”, as ‘My lunch, such as bread, soup, croquette, milk’. Drinks were described as ‘My drinks, such as coffee, tea, soft drinks’ and snacks were described as ‘My snacks, such as cookies, fruit, candy and chips’. Answer categories were: entirely bought at school (in the canteen or from vending machines), partly bought at school (in the canteen or from a vending machine) or entirely taken from home or bought somewhere else. The frequency of purchase was also measured, but data for these measures will be reported elsewhere. Several filler questions were included (e.g., ‘the service in this canteen is good’), which are not reported.

### 2.6. Data Analysis

Outcome measures were analysed using regression approaches as appropriate for the type of measure, with the location of the canteen (the two schools) and period (1–4) and the location of canteen*period interaction as independent variables. All variables were dummy coded and for the period selected such that contrasts could be specified. 

For hypothesis 1, two specified contrasts compared changes to a higher percentage healthier products separately for Ede and Velp with “type of product” (either a ‘healthier choice’ coded 0 or an ‘unhealthy choice’ coded 1) as the dependent variable in binary logistic regression analyses (contrast A and B).

Contrast was also specified to test the second hypothesis that a gradual change in the Velp canteen would lead to more healthier food choices than a more abrupt change in the Ede canteen. This contrast that was included as a dummy in the analysis compared the gradual change in the Velp canteen (from 60% to 70% to 80%) to the abrupt change of 60% to 80% in the Ede canteen, again with the dependent variable ‘type of product’ which could be either an unhealthy choice or a healthier choice product (contrast C). In total seven orthogonal dummy variables were included of which one compared the contrast of interest. 

Only completed online questionnaires by students were included in the dataset. For the other outcome measures from the surveys, we use identical dummy regression approaches but with continuous dependent variables (e.g., satisfaction). For outlet choice, the answer categories ‘entirely bought at school’ and ‘partly bought at school’ were combined to simplify interpretation. To account for multiple comparisons, a lower significance level of *p* ≤ 0.01 was used in all analyses. The data of this manuscript will be made available on https://zenodo.org/ by the authors.

## 3. Results

### 3.1. Descriptives

#### 3.1.1. Canteen Choice Characteristics

Table 2 shows an overview of the product choice across the different canteens and periods. Overall, the dataset contains 114,761 unique transactions and 198,053 products sold (i.e., 1.7 products per transaction). In the entire period of 10 months, the canteen in Ede sold 134,834 products and the canteen in Velp sold 63,219 products. This indicates that about two-thirds of all products were sold in Ede. There is a strong fluctuation in sold products in both canteens. In Ede, the canteen sold in total 34,739, 22,319, 64,747, and 13,029 products in period 1, 2, 3, and 4, respectively. In Velp this was 18,101, 17,184, 23,633 and 4301 products in total in each period, respectively.

The share of product groups in total sales varied largely. The four largest product groups in terms of the total number of products sold make up for 79% of the sales. In total across both canteens, 77.690 sandwiches were sold (39.2% of all products sold). As such, this is the largest product group in terms of sales. The second-largest product group in terms of sales are beverages. A total of 43.259 beverages were sold (21.8% of all products sold). The third-largest product group is ‘savoury snacks’ which represents 11.2% of all sales (22.235 snacks). The fourth-largest product group is ‘sweet snacks’ (6.4% of sales; 12.692 snacks). Throughout the year, in total 2.754 pieces of fresh fruit and snack vegetables were sold in the two canteens. This represents 1.4% of all sold products (2.754 out of 198.053). Across both canteens, 10.462 portions of soup were sold (5.3%) and 10.565 dairy products (5.3%).

#### 3.1.2. Differences between Product Groups

There are substantial differences in the share of healthier choice sales across product groups. Snack vegetables and fruit are all ‘healthier choice’ products, hence 100% of all sales are healthy. Figure 1 displays the share of ‘healthier choices’ sold per product group for five important product groups (i.e., sandwiches, beverages, savoury snacks, sweet snacks and soups). Of all sold savoury snacks, 1.2% on average is healthier (i.e., unsalted nuts). 

#### 3.1.3. Spending and Share of Healthier Products per Transaction

Per unique transaction, customers bought on average for 3.08 EUR (SD = 1.96). Customers spent significantly more per transaction in the periods in which 80% of the assortment was healthy. In Ede, unique transactions were on average €2.86 (SD = 1.58) in the first period with a 60% healthier assortment, while during the 80% healthier assortment periods 2 to 4, on average €3.29 (SD = 2.15) was spent per transaction; contrast estimate = 0.34, SE = 0.02, F (1,114753), = 198.9, *p* < 0.001. In Velp, a similar pattern was observed. Unique transactions were, on average, €2.74 (SD = 1.53) in the first period with a 60% healthier assortment, while during the 80% healthier assortment periods 3 and 4, on average €3.08 (SD = 2.00) was spent per transaction; contrast estimate = 0.14, SE = 0.03, F (1,114753), = 27.82, *p* < 0.001. The share of healthier choices per transaction was on average 31.4%. Of all transactions, 58.4% contained no healthier options and 22% contained only healthier options. 

#### 3.1.4. Student Survey

In total, 1564 students (54% women) completed the questionnaire: 892 in Ede and 672 in Velp. The number of individuals that filled in the questionnaire was 416 in period 1 (256 in Ede and 160 in Velp), 397 in period 2 (217 in Ede and 180 in Velp), 424 in period 3 (249 in Ede and 175 in Velp) and 327 in period 4 (170 in Ede and 157 in Velp).

In the total sample, the mean age of respondents was 18.6 years (SD = 2.5; range 15–50, median 18). Respondents in Ede were somewhat older (M = 18.8, SD = 2.7) than respondents in Velp (M = 18.3, SD = 2.1); t (1562) = 3.65, *p* < 0.001. On average, students reported being present at school for almost the entire school week (M = 4.4 days, SD = 1.0). Respondents followed a wide variety of educational programmes (Table 3). 

### 3.2. Hypotheses Testing

#### 3.2.1. Hypothesis 1: Increased Presence of Healthier Products Leads to Increased Sales of Healthier Products

Across the entire study period and both canteens, 32.8% of all sold products were ‘healthier choice’ products. Table 4 includes the percentage of sales of ‘healthier choice’ products together with other key outcome measures. Across canteens, 31.1% of all products sold were a healthier choice in period 1 and this increased to 35.9% in period 4. We hypothesized (Hypothesis 1) that a larger visible proportion of healthier products will lead to significantly more sales of these products relative to unhealthy products. The four periods in the Ede and Velp canteen were separately contrasted.

For Ede, a binary logistic regression was conducted with a dummy coded variable that specified the contrast between the first 60% period and the change to three 80% periods with the type of choice (‘healthier’ or ‘unhealthy choice’) as the dependent variable (contrast A). As predicted, the dummy variable significantly predicted type of choice (estimated logistic regression coefficient *β* = −0.08, Wald = 31.87, *p* < 0.001). For Velp, a binary logistic regression was conducted with a dummy coded variable that specified the contrast between the first 60% period, the second 70% period and the change to two 80% periods with the type of choice (‘healthier’ or ‘unhealthy choice’) as the dependent variable (contrast B). There was no significant difference between the first 60% period and the second 70% period (estimated logistic regression coefficient *β* =.01, Wald = 0.20, *p* = 0.65). There was a significant difference between the last two periods 3 and 4 and the first 60% period (estimated logistic regression coefficient *β* = −0.28, Wald = 187.07, *p* < 0.001). Hence, hypothesis 1 is confirmed (Table 5). 

#### 3.2.2. Hypothesis 2: Gradual Increase in Healthier Products Leads to Healthier Choices Than an Abrupt Increase

We furthermore hypothesized (Hypothesis 2) that a slow change towards a healthier assortment (Velp) would lead to higher sales of ‘healthier choice’ products in 80% healthier choice periods than an abrupt change in the assortment (Ede). Binary logistic regression was conducted with a dummy coded variable that specified the contrast between the change to 80% in Ede and Velp and type of choice (‘healthier’ or ‘unhealthy choice’) as the dependent variable (contrast C, Table 6). As predicted, the dummy variable significantly predicted type of choice (estimated logistic regression coefficient *β* = −0.08, Wald = 68.27, *p* < 0.001). As can be seen in Table 4 and Figure 2, the share of healthier choices is higher in period 3 and 4 in Velp compared to period 2 to 4 in Ede. Hence, hypothesis 2 is confirmed. This shows that (a) indeed a larger visible proportion of healthier products will lead to significantly more sales of these products relative to unhealthy products, and (b) that, indeed, a slow change towards a healthier assortment (Velp) would lead to higher sales of ‘healthier choice’ products in 80% healthier choice periods than an abrupt change in the assortment (Ede).

### 3.3. Additional Measures

For satisfaction, scores were in the mid-range of the scale. Table 4 also shows that student satisfaction is insensitive to assortment changes over time. During the year, students also felt equally free to choose and felt that they can eat healthily. 

Overall, large groups of students purchased their lunch (47.6%), drinks (61.9%) and snacks (64.1%) somewhere else or brought it from home. Only a small group of students purchased their lunch (8.7%), drinks (8.3%) and snacks (7.7%) entirely at school. For the small group of students that purchased their entire lunch at school, no significant changes in student satisfaction rates occurred over time (all *p*-values > 0.12). Many students combined purchases in the canteen with purchases elsewhere or home brought foods and drinks (lunch 43.7%, drinks 29.8%, snacks 28.2%). Over time, small and significant changes occurred in where students purchase their products, but the pattern is not consistent (Table 5 and Table 6). 

## 4. Discussion

This study aimed to understand the impact of increasing the visible healthfulness of the assortment of vocational school canteens on product sales and student attitudes. Overall, sales of healthier products increased from 31% in the baseline period to almost 36% in the final period of the school year. This relative small improvement corresponds to findings of other canteen or retail interventions in various settings. For example, a randomized controlled trial in worksite cafeterias in the Netherlands showed similar increased sales figures, particularly for sandwiches, cheese and fruit [28]. A recent intervention study in worksite cafeterias in the United Kingdom in which the proportion of healthier foods available was increased showed a reduction of almost 7% in energy purchased from targeted food products [29]. 

School canteen sales were driven by sandwiches, beverages, and savoury snacks. About half of the displayed beverages of the canteen were healthier options with less or no sugar. This percentage was mirrored in sales (45.3% healthier in the abrupt changing canteen, 53% in gradually changing canteen). Throughout the study period, the share of sales of healthier beverages increased from 42% to 51%. This shows that increasing the relative share of displayed healthier products can make a difference, but particularly in situations where consumers are quite indifferent in their choices between available options. It would, therefore, be good to promote healthier options in precisely these product groups or reformulate the composition or recipe of products to improve their healthiness. Relatively small changes can be acceptable to consumers. For example, a study in which the default bread type of a sandwich was manipulated (whole grain versus white) showed that consumers were equally satisfied with both bread types [30]. 

We expected a gradual increase to be more acceptable than an abrupt increase. This expectation was confirmed by the sales data. Across product categories, a gradual change led to a higher share of healthier products sold than an abrupt change. This pattern of larger increases in the share of healthier products sold was observed in beverages, dairy and salads and less so in sandwiches. Questions that arise in this respect are what other changes in purchase patterns take place and whether there is a threshold for change to happen in purchase behaviour. For example, students spent a bit more money in the canteen during the 80% healthier products periods, but further analyses could look at why this is the case. It might be that there are spill-over effects in that students spend more money to reward themselves when purchasing healthy products (see [31], for a study in a retail setting).

Research on psychological reactance suggests that people may respond negatively to restrictions in the choice of their preferred products [32]. Our study showed that satisfaction with the canteen and students’ perceived freedom in choice was not affected by the intervention. Important to note is that choices were not restricted and unhealthy products that were visibly low in availability could be refilled when they were (almost) sold out. 

Many intervention studies focused on increasing the consumption of fruit and vegetables in the school setting [33]. Only 1% of revenues were generated from fruit and vegetable sales, showing that fresh fruit and vegetables are no popular choices in school canteens without school meal systems, even when increasing their availability. This may be because students feel that they can easily bring fruits from home and tend to purchase foods as a supplement to a packed or somewhere else bought lunch. Similarly, in some product groups, such as soups and salads, the healthier option is an attractive option for customers, as most of the sales were from this group. Nevertheless, total sales of these products are relatively low. 

A strength of the current field study is the use of unique check out data for a longer period and careful monitoring of the changes in this period. Additionally, a large group of students was surveyed throughout the year which gives insights in intervention acceptance and changes in attitudes. Nevertheless, there are a few limitations of the study which need to be mentioned. The canteen scan to determine how many healthy products were on display was only a snapshot in time. The share of healthier products displayed likely varied not only within the day but also across days. Reliable vending machine sales data was not available due to data collection errors. This type of data could have shown potential compensation effects inside the school. Moreover, single items were used in the survey to measure constructs such as satisfaction which may have limited reliability. 

Important to note is that employees also purchased products in the canteen and only a small group of students in this study purchased their lunch (9%), drinks (8%) and snacks (8%) entirely at school. This is typical for many canteens at secondary and vocational schools in the Netherlands and partly driven by high price perceptions of the foods sold [34]. Consequently, canteens often do their very best to attract customers to get sufficient turnover. In such a situation without guaranteed customers, it is relevant to see the impact of healthier school assortment policies. Nevertheless, this also means that results cannot fully be extended to other situations and countries. Furthermore, an important limitation is that there were no control canteens in which there were no guidelines on the type and size of the offered assortment. Moreover, different types of students in terms of type of education were present at the two school locations. This study shows what happens when basic entry levels of 60% healthier assortment are exceeded. This limits the generalization to other canteens at vocational or secondary schools without school canteen programs. However, the healthiness level of baseline condition of the two canteens in this study is relatively easy to achieve. 

## 5. Conclusions

Results suggest that increasing the availability of healthier products in school canteens leads to small positive changes in sales of these products, particularly in the product groups beverages and sandwiches. A gradual introduction may ensure that students slowly get used to assortment changes. Our findings have relevant policy implications. It is repeatedly stressed by various influential public policy researchers that there is a need for a positive and enabling food environment for young people (‘culture of health’) [35]. Our results show the potential impact of an environmental intervention that is feasible to apply in various settings such as canteens in schools, companies and restaurants. Although the effects are relatively small, over time the share of healthier products sold may increase by making healthier options more appealing and shifting social norms of what is preferred to eat. As such, this nudge type intervention to encourage healthier choices could complement other policy instruments such as laws, taxes, and education as part of an integral approach to change behaviour. 

## Figures and Tables

**Figure 1 ijerph-17-04352-f001:**
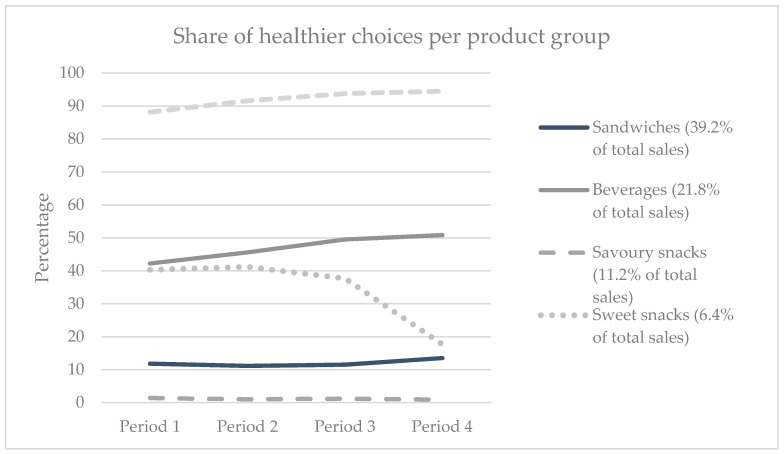
Share of healthier products across four time periods for a selection of product groups.

**Figure 2 ijerph-17-04352-f002:**
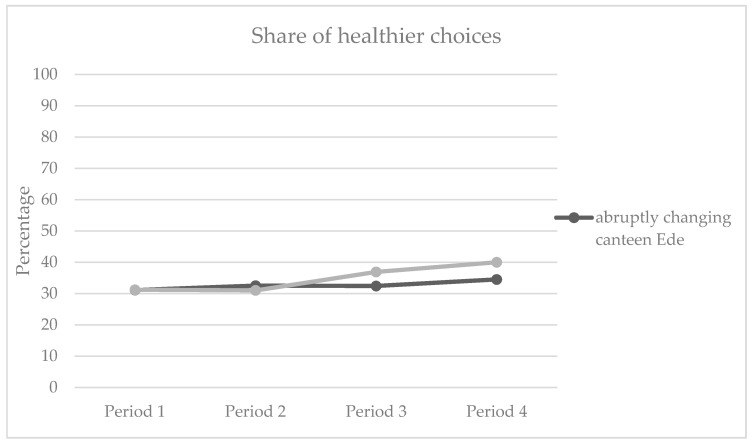
Share of healthier products sold across four time periods in Ede and Velp.

**Table 1 ijerph-17-04352-t001:** Experimental design of the study in two canteens.

Canteen	Academic Year (September til June)			
	Baseline period 1 (about two months)	Experimental period 2 (about two months)	Experimental period 3 (about 3–4 months)	Experimental period 4 (two months) *
Canteen Ede	60% Healthier Choice products in the assortment	80% Healthier Choice products in the assortment	80% Healthier Choice products in the assortment	80% Healthier Choice products in the assortment
Canteen Velp	60% Healthier Choice products in the assortment	70% Healthier Choice products in the assortment	80% Healthier Choice products in the assortment	80% Healthier Choice products in the assortment

* Initially a washout period was planned without adhering to the 80% rule, but the catering company was willing to continue with this proportion of healthier choices in the assortment.

**Table 2 ijerph-17-04352-t002:** Descriptives of experimental design, sales, revenues and number of surveyed participants in each of the four periods in the two canteens.

	Canteen Ede (Fast Change to 80% Healthier Choice Products)				Canteen Velp (Slow Change to 80% Healthier Choice Products)			
	Period 1 (12 September–18 November)	Period 2 (19 November–15 January)	Period 3 (16 January–30 April)	Period 4 (1 May–1 July)	Period 1 (12 September–13 November)	Period 2 (14 November–15 January)	Period 3 (16 January–30 April)	Period 4 (1 May–1 July)
# surveyed participants	256	217	249	170	160	180	175	157
% healthier choices displayed	60%	80%	80%	80%	60%	70%	80%	80%
Number of weeks open	8	6	13	5	9	7	13	5
Average total number of products sold per week	4342	3720	4982	2606	2011	2455	1818	860
% Healthier Choice products sold of total sales (total number sold) ^1^								
Sandwiches	12.1% (1704/14,122)	10.8% (978/9083)	10.5% (2769/26,426)	12.5% (646/5171)	11.2% (736/6561)	11.5% (762/6655)	14.6% (1200/8226)	19.6% (284/1446)
Beverages	40.5% (3027/7467)	46.2% (2188/4740)	48.1% (6858/14,244)	46.5% (1399/3010)	45.2% (1859/4114)	44.8% (1616/3607)	59.5% (2956/4967)	62.4% (693/1110)
Savoury snacks	1.4% (57/4013)	0.9% (22/2474)	1.3% (100/7995)	0.7% (11/1500)	1.3% (23/1816)	1.1% (19/1773)	1.0% (23/2265)	1.8% (7/399)
Sweet snacks	34.9% (751/2149)	39.4% (467/1186)	37.1% (1055/2845)	15.0% (68/452)	47.9% (720/1502)	42.8% (580/1354)	38.3% (1108/2894)	21.6% (67/310)
Soups	96.9% (1421/1467)	94.3% (1164/1235)	97.3% (3028/3112)	98.0% (595/607)	74.3% (681/916)	88.5% (954/1078)	87.6% (1529/1746)	87.4% (263/301)
Dairy	36.0% (644/1789)	30.9% (305/986)	36.1% (974/2696)	48.6% (297/611)	23.3% (375/1609)	20.7 (206/995)	20.0 (301/1507)	27.4 (102/372)
Salads	92.7% (662/714)	87.8% (332/378)	94.3% (1185/1256)	97.6% (327/335)	92.4% (194/210)	78.7% (129/164)	55.6% ^2^ (50/90)	100% (44/44)
Sauces	95.7% (1942/2029)	93.2% (1417/1521)	99.2% (3845/3877)	100% (767/767)	88.1% (922/1047)	82.4% (842/1022)	95.9% (1178/1229)	100% (178/178)
Fruits and vegetables	100% (443/443)	100% (318/318)	100% (952/952)	100% (321/321)	100% (112/112)	100% (180/180)	100% (346/346)	100% (82/82)
Bread—no topping	23.8% (127/534)	14.2% (56/394)	16.2% (218/1344)	26.4% (66/250)	9.0% (19/210)	9.3% (33/356)	11.3% (41/363)	0% (0/59)
Revenues per transaction	2.86 (1.58)	2.70 (1.53)	3.67 (2.43)	2.88 (1.54)	2.74(1.53)	2.80 (1.84)	3.20(2.11)	2.56 (1.31)
Products sold per transaction	1.56 (0.85)	1.49 (0.81)	2.01 (1.33)	1.58 (0.84)	1.59 (0.91)	1.61 (1.01)	1.95 (1.31)	1.52 (0.79)

^1^ Separate sweet and savoury bread toppings had very low sales over the entire period (25 in total) so they are omitted from this table. ^2^ In period 3, only 55.6% of all salads sold were healthy, which is considerably lower than in other months. The canteen audit scan showed that this may result from the lack of offering salads in this time periods or only unhealthy salads.

**Table 3 ijerph-17-04352-t003:** Characteristics of survey participants across the two canteen locations.

Characteristic	Ede	Velp	Total Sample
Gender			
Male	434	282	716
Female	458	390	848
Total	892	672	1564
Age (mean, SD)	18.8 (2.7)	18.3 (2.1)	18.6 (2.5)
Average number of days present at school	4.4 (1.1)	4.5 (0.9)	4.4 (1.0)
Type of education followed			
Business, Economics and Administration	246	175	421
Media, ICT, Image & Sound	294	121	365
Care and Wellbeing	148	204	352
Catering, Tourism, Recreation & Wellness	60	69	126
Another type of education	144	103	247

**Table 4 ijerph-17-04352-t004:** Outcome measures of cash register and survey data in each of the four periods in two canteens.

	Canteen Ede (Fast Change to 80% ‘Healthier Choice’ Products)				Canteen Velp (Slow Change to 80% ‘Healthier Choice’ Products)			
	Period 1 (12 September–18 November)	Period 2 (19 November–15 January)	Period 3 (16 January–30 April)	Period 4 (1 May–1 July)	Period 1 (12 September–13 November)	Period 2 (14 November–15 January)	Period 3 (16 January–30 April)	Period 4 (1 May–1 July)
**Cash Register Data**								
Relative proportion of Healthier Choice products sold of total sales (%)	31.1%	32.5%	32.4%	34.5%	31.2%	31.0%	36.9%	40.0%
Relative proportion of Healthier Choice products sold per transaction (%)	29.9%	31.0%	31.0%	33.1%	29.7%	30.2%	35.5%	39.7%
**Student Satisfaction**								
I am satisfied about this canteen *	3.2 (0.9)	3.2 (0.8)	3.1 (0.8)	3.0 (0.8)	2.9 (0.9)	3.1 (1.0)	2.9 (0.8)	3.0 (0.9)
In this canteen you can eat healthily *	3.4 (0.8)	3.4 (0.8)	3.3 (0.8)	3.2 (0.7)	3.2 (0.8)	3.3 (0.8)	3.2 (0.7)	3.2 (0.8)
I feel free to choose what I like *	3.6 (0.9)	3.6 (0.9)	3.6 (0.9)	3.4 (0.9)	3.4 (1.0)	3.6 (1.0)	3.3 (0.9)	3.6 (1.0)
**Outlet Choice**								
Where do you get your lunch?								
Entirely bought at school	18.8%	8.3%	10.4%	5.9%	1.9%	7.2%	7.4%	3.2%
Partly bought at school	44.1%	61.3%	43.4%	50.0%	28.7%	40.0%	29.1%	48.4%
Entirely from somewhere else	37.1%	30.4%	46.2%	44.1%	69.4%	52.8%	63.4%	48.4%
Where do you get your drinks?								
Entirely bought at school	14.5%	7.4%	10.0%	7.1%	2.5%	7.8%	8.0%	5.1%
Partly bought at school	28.5%	39.2%	27.7%	38.2%	18.1%	28.9%	25.1%	31.2%
Entirely from somewhere else	57.0%	53.5%	62.2%	54.7%	79.4%	63.3%	66.9%	63.7%
Where do you get your snacks?								
Entirely bought at school	13.7%	6.9%	9.2%	5.9%	3.1%	7.2%	6.3%	5.1%
Partly bought at school	31.6%	26.3%	26.5%	34.7%	18.8%	32.2%	22.9%	31.8%
Entirely from somewhere else	54.7%	66.8%	64.3%	59.4%	78.1%	60.6%	70.9%	63.1%

* All items were measures on five-point scales ranging from 1 ‘totally disagree’ to 5 ‘totally agree’.

**Table 5 ijerph-17-04352-t005:** Regression results hypothesis 1 and additional measures.

	Unstandardized Intercept B0	Unstandardized B (Dummy Contrast)	Fit
**Cash register data** (**H1**)			
Ede (60 –> 80% period change)			
Healthy choice	0.80	−0.08 *	31.87 ^a^
Velp (60 –> 70% period change)			
Healthy choice	0.79	0.01	0.20 ^a^
Velp (60 –> 80% period change)			
Healthy choice	0.79	−0.28 *	187.07 ^a^
**Student satisfaction** (**additional measure**)			
Ede (60 –> 80% period change)			
Satisfaction	3.05	0.10	0.02 ^b^
Can eat healthily	3.28	0.07	0.01 ^b^
Free to choose	3.52	0.07	0.01 ^b^
Velp (60 –> 70% –> 80% period change)			
Satisfaction	3.05	−0.04	0.02 ^b^
Can eat healthily	3.28	−0.03	0.01 ^b^
Free to choose	3.52	−0.03	0.01 ^b^
**Customer patronage** (**additional measure**)			
Ede (60 –> 80% period change)			
Lunch (school purchase)	−0.04	−0.35 *	7.39 ^a^
Drinks (school purchase)	0.53	−0.35 *	6.78 ^a^
Snacks (school purchase)	0.62	−0.47 *	11.82 ^a^
Velp (60 –> 70% –> 80% period change)			
Lunch (school purchase)	−0.04	0.14 *	7.39 ^a^
Drinks (school purchase)	0.53	0.14 *	6.78 ^a^
Snacks (school purchase)	0.62	0.19 *	11.82 ^a^

* *p* ≤ 0.01 ^a^ Wald ^b^ Adjusted R^2^.

**Table 6 ijerph-17-04352-t006:** Regression results hypothesis 2 and additional measures.

	Unstandardized Intercept B0	Unstandardized B (Dummy Contrast)	Fit
**Cash register data** (**H2**)			
Healthy choice	0.68 *	−0.08 *	0.003 ^b^ 31.87 ^a^
**Student evaluation** (**additional measure**)			
Satisfaction	3.05	0.01	0.02 ^b^
Can eat healthily	3.28	0.01	0.01 ^b^
Free to choose	3.52	0.01	0.01 ^b^
**Customer patronage** (**additional measure**)			
Lunch (school purchase)	−0.04	−0.04 *	7.39 ^a^
Drinks (school purchase)	0.53	−0.04 *	6.78 ^a^
Snacks (school purchase)	0.62	−0.06 *	11.82 ^a^

* *p* ≤ 0.01 ^a^ Wald ^b^ Adjusted R^2^.

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
