# Peer review of "The Impact of a Gradual Healthier Assortment among Vocational Schools Participating in a School Canteen Programme: Evidence from Sales and Student Survey Data"

_ijerph, 2020, doi:10.3390/ijerph17124352_

Round 1
Reviewer 1 Report
This paper examines the impact of gradual healthier products availability in vocational school canteens that follow a healthy canteen school program. Authors find that increasing the visible share of healthier products leads to small changes in sales of these products. The goals and rationale of the study are clearly stated in the paper. Also, measures used to collect data and data collection procedure are well described. The presentation of the results can be improved.
Please find below my concerns that would need further clarifications.
- The title of the article suggests a broader topic than presented in the article. Both vocational schools participating in the study conform to the guidelines of the healthy canteen school program and have a minimum of 60% healthier choice products in the visible assortment at the baseline condition. This percentage is much larger than the average percentage of healthier products in other vocational schools without school canteen programs. Thus, you might want to include in the title that this study focuses on schools that follow nutrition programs.
- Lines 26-54. I feel a lack of a review of the literature in the field. School eating behavior and school food environment interventions has been largely investigated the last decades.
- Line 159. You mention that there were weekly price promotions, both for healthier and unhealthier products. Were promotions balanced between healthy and unhealthy products and across all product categories in order no price effects have affect your results? This arises questions whether the outcome measures were solely affected by the increased visible share of healthier products.
- There are some discrepancies when referring to your time periods throughout the manuscript:
First, be consistent with the period names. In the first part of methods section and in Table 1, you mention a “baseline period” and periods “1,2,3”. In Tables 5 & 6 as well as in the rest of the manuscript, “the baseline period” has been named “period 1” and treatments are named 2,3,4. This confuses the reader.
Second, the length of the last two periods are depicted differently among tables 1 (2 months and 4 months respectively), and table 5 (3.5 months and 2 months respectively). Please check which is the right timeline and correct it.
- Lines 202-203/ 206-207. Correct the repetition of the measurement method.
- Line 222 – 241. If you present the forms of the model specification you estimated in data analysis, the regression approaches you followed will become clearer to the reader.
- Lines 262-263. In this paragraph (and generally in the descriptive section) there are only dry numbers which are also in the tables/figures and you might want to comment on. Why do you think customers spend more per transaction in the 80% periods? Was it because they purchased more products per transaction? In period 3 in both schools we see an increase in the number of products per transaction, but this is not consistent in the rest periods. Could have been the case of purchasing a healthy product and an unhealthy product as a personal “reward”/compensation effect? Or did they spent more per transaction because they bought more/only healthy products which are probably more expensive than unhealthy products?
- Lines 276-278. There is a discrepancy between the results in the manuscript and results depicted in Figure 1. Are sauces one of the four product groups with the highest number of total sales? If yes, it should have been included in the figure.
- Line 289. In sample characteristics table you present the educational programs. Why do you include this in table 2 since you do not use it anywhere? Why do you think is important to know that? Did you detect different canteen satisfaction results based on the educational program followed, e.g. for student in Catering, Tourism, Recreation & Wellness who might be more involved with food issues. Did you find any correlation between the healthier shares and the educational programs followed?
- As far as the sample’s age concerns, isn’t it a bit on the edge to be called adolescents (Ede=18.8 & Velp=18.3)? It is not my field of knowledge, but WHO defines “Adolescents” as individuals in the 10-19 years age group and “Youth” as the 15-24 years age group. What is your sample’s age range?
- Line 305. Does this mean that 80% is a threshold for a change in students eating behavior for vocational schools following a nutrition program? At 70% period it seems that there is a small decrease in healthier products sold.
- Line 338. I suggest that authors consider for students that entirely bough lunch at school, examining whether any differences in their canteen satisfaction measures exist among the 4 periods?
Author Response
We want to thank reviewer for taking the time to review our paper. We appreciate the positive evaluation of our paper. The paper has been revised in response to the reviewers’ comments and we feel that the paper has benefitted greatly from the suggestions. We made several changes to improve the presentation of results. For example, we changed the sequence of tables to make the flow more logical and improved the lay-out of figure 1 and 2. Please find our specific response below.
Reviewer 1
This paper examines the impact of gradual healthier products availability in vocational school canteens that follow a healthy canteen school program. Authors find that increasing the visible share of healthier products leads to small changes in sales of these products. The goals and rationale of the study are clearly stated in the paper. Also, measures used to collect data and data collection procedure are well described. The presentation of the results can be improved.
Please find below my concerns that would need further clarifications.
- The title of the article suggests a broader topic than presented in the article. Both vocational schools participating in the study conform to the guidelines of the healthy canteen school program and have a minimum of 60% healthier choice products in the visible assortment at the baseline condition. This percentage is much larger than the average percentage of healthier products in other vocational schools without school canteen programs. Thus, you might want to include in the title that this study focuses on schools that follow nutrition programs.
Response 2: We changed the title into: ‘The impact of a gradual healthier assortment in a school canteen programme: Evidence from sales and survey data.’
- Lines 26-54. I feel a lack of a review of the literature in the field. School eating behavior and school food environment interventions has been largely investigated the last decades.
Response 3: Thank you this suggestion. It is indeed correct that our literature review was limited. Recently, two important systematic review papers were published which clearly give an overview of school food environment interventions. We therefore decided to add these to the introduction to summarize key findings in the past decade (line 47-54): ‘A recent meta-analysis of 91 school food environment policies interventions (55 in US/Canada, 36 in Europe/New Zealand) showed efficacy of a range of food environment policies, including direct provision, competitive food/beverage standards, and school meal standards [35]. Another review looked at 25 studies reporting results of implementing behavioral nudges in school canteen to encourage healthier eating. These nudges aimed to make healthier choices more convenient, attractive and acceptable, such as changing the order of serving vegetables or prompting children to take a fruit. Results of the majority of studies showed an increase in the selection or consumption of target foods such as fruit [34].’
- Line 159. You mention that there were weekly price promotions, both for healthier and unhealthier products. Were promotions balanced between healthy and unhealthy products and across all product categories in order no price effects have affect your results? This arises questions whether the outcome measures were solely affected by the increased visible share of healthier products.
Response 4: To increase realism of the study, the regular price promotions of the caterer were kept in place, which sometimes included healthier promotions and sometimes unhealthier promotions. Important is that promotions were similar across canteens. Hence, price promotions were no variable in our design and differences in sales across both canteens due to the promotion do not impact results. We now added to line 173: ‘Price promotions were similar in both canteens.’
- There are some discrepancies when referring to your time periods throughout the manuscript:
First, be consistent with the period names. In the first part of methods section and in Table 1, you mention a “baseline period” and periods “1,2,3”. In Tables 5 & 6 as well as in the rest of the manuscript, “the baseline period” has been named “period 1” and treatments are named 2,3,4. This confuses the reader.
Second, the length of the last two periods are depicted differently among tables 1 (2 months and 4 months respectively), and table 5 (3.5 months and 2 months respectively). Please check which is the right timeline and correct it.
Response 5: We apologize for this inconsistency. It is indeed confusing. We now call the baseline period; period 1. We also corrected the mistake in timeline. The final period 4 was about 2 months.
We adapted table 1 and also added the following sentence to the methodology section (line 162), given that table 3 and 4 give the correct dates: ‘The exact time of change to another condition was done in agreement with canteen personnel, so that enough products could be purchased.’
- Lines 202-203/ 206-207. Correct the repetition of the measurement method.
Response 6: We changed the period indication in sentence 199.
- Line 222 – 241. If you present the forms of the model specification you estimated in data analysis, the regression approaches you followed will become clearer to the reader.
Response 7: We now indicate more clearly which specified contrast is used in for which analysis/hypothesis. We did that by including contrast A, B and C (see data analysis section 2.6 and the results section).
- Lines 262-263. In this paragraph (and generally in the descriptive section) there are only dry numbers which are also in the tables/figures and you might want to comment on. Why do you think customers spend more per transaction in the 80% periods? Was it because they purchased more products per transaction? In period 3 in both schools we see an increase in the number of products per transaction, but this is not consistent in the rest periods. Could have been the case of purchasing a healthy product and an unhealthy product as a personal “reward”/compensation effect? Or did they spent more per transaction because they bought more/only healthy products which are probably more expensive than unhealthy products?
Response 8: These are relevant questions that arise when reading the results. To prevent speculation in the result section, we added a paragraph in the discussion section (line 402): ‘Questions that arise in this respect are what other changes in purchase patterns take place and whether there is a threshold for change to happen in purchase behaviour. For example, students spent a bit more money in the canteen during the 80% healthier products periods, but further analyses could look at why this is the case. It might be that there are spill-over effects in that students spend more money to reward themselves when purchasing healthy products [see 33, for a study in a retail setting].’
- Lines 276-278. There is a discrepancy between the results in the manuscript and results depicted in Figure 1. Are sauces one of the four product groups with the highest number of total sales? If yes, it should have been included in the figure.
Response 9: In total across the two canteens, 11.670 sauces were sold, which is 5.9% of all products sold. This is not the fourth-highest group. The fourth-highest group is sweet snacks (12.692 sweet snacks; 6.4% of sales). We changed this in the text at line 275 and line 282-283. We also changed figure 1 which now includes five interesting and important product groups (i.e. sandwiches, beverages, savoury snacks, sweet snacks and soups). We also indicated in figure 1 what the contribution to total sales is for each of these product groups, so that the reader gets an indication of their importance.
- Line 289. In sample characteristics table you present the educational programs. Why do you include this in table 2 since you do not use it anywhere? Why do you think is important to know that? Did you detect different canteen satisfaction results based on the educational program followed, e.g. for student in Catering, Tourism, Recreation & Wellness who might be more involved with food issues. Did you find any correlation between the healthier shares and the educational programs followed?
Response 10: We included it so that it gives a description of the background and possible interests of the student population. We did not expect strong differences. The groups are relatively small and different in size to justify subgroup analysis. Unfortunately, we cannot calculate correlations as data on healthier shares was not collected on an individual level, but at transaction level. So, it is not possible to connect the two datasets.
However, as also indicated by reviewer 2 (response 21), different educational program students may have different health and nutrition related interests, which may have impacted the results. We included a sentence in the part of the discussion (line 436-439): ‘Furthermore, an important limitation is that there were no control canteens in which there were no guidelines on the type and size of the offered assortment. Moreover, different types of students in terms of type of education were present at the two school locations. This study shows what happens when basic entry levels of 60% healthier assortment are exceeded’.
- As far as the sample’s age concerns, isn’t it a bit on the edge to be called adolescents (Ede=18.8 & Velp=18.3)? It is not my field of knowledge, but WHO defines “Adolescents” as individuals in the 10-19 years age group and “Youth” as the 15-24 years age group. What is your sample’s age range?
Response 11: The age range of the surveyed sample is 15 to 50 year old. This one person of 50 years old is an outlier, so we also added the median to the text (line 304): ‘In the total sample, the mean age of respondents was 18.6 years (SD=2.5; range 15-50, median 18).’
It is indeed generally late adolescence or youth. That is why we call them students in the title and throughout the paper. We did add ‘young adults’ to one sentence in the introduction (line 31): ‘Schools are increasingly seen as an ideal setting for implementing interventions to make young people eat and drink healthier as the majority of children, adolescents and young adults can be reached for longer periods and up to 50% of daily caloric intake takes place at school [3].’
- Line 305. Does this mean that 80% is a threshold for a change in students eating behavior for vocational schools following a nutrition program? At 70% period it seems that there is a small decrease in healthier products sold.
Response 12: We feel that more research is needed to understand this interesting question. We added a paragraph to line 402, see also response 8: ‘Questions that arise in this respect are what other changes in purchase patterns take place and whether there is a threshold for change to happen in purchase behaviour. For example, students spent a bit more money in the canteen during the 80% healthier products periods, but further analyses could look at why this is the case. It might be that there are spill-over effects in that students spend more money to reward themselves when purchasing healthy products [see 33, for a study in a retail setting].’
- Line 338. I suggest that authors consider for students that entirely bough lunch at school, examining whether any differences in their canteen satisfaction measures exist among the 4 periods?
Response 13: Thank you for this interesting suggestion. We made these calculations for the four time periods, only for students that entirely bought lunch at school. We now report this at page 9 (line 363); ‘For the small group of students that purchased their entire lunch at school, no significant changes in student satisfaction rates occurred over time (all P-values > 0.12).’
Reviewer 2 Report
- Page 2, line 60-- ... recommend levels of healthy relative to unhealthy foods on display from..
Can authors expand to include the definitions for what determining the "healthy" and "unhealthy" foods?
- Page 3, line 103 -- This study was conducted from September 2016 until July 2017 at two vocational school canteens
This noted study period is inconsistent with the period noted in Table 1 (September till June - 10 months). Do authors consider "seasonality" issue for fresh produce availability? Was this an issue at that geographic location?
- Page 3, line 107 -- Instead, students bring their own packed lunch or purchase meals, snacks and drinks at school or at food outlets nearby school...
Would authors be able to comment on the price differences between the food purchased from school vs. food purchased from food outlets nearby schools? Would this influence on food purchasing behavior?
- Page 3, line 126 -- ... when they are healthy or reasonably healthy...
Can authors provide some examples for those food category?
- Page 4, line 146 -- In the first baseline phase (September until November 2017), the visible...
Do authors mean "2016"?
- Page 4, line 150 and page 5 line 169 -- ... when their stock is too low../ ... on display was too low, ....
Can authors provide the definition of "too low"? Was there any issue on supplying healthy or unhealthy food at school?
- Page 6, line 272 -- Differences between product groups
Can this paragraph be moved to the section "Canteen choice characteristics" (staring line 244)?
- Page 7, table 2 -- Type of education of "Care and Wellbeing" followed for school Ede and Velp are 148 and 204, respectively.
Do authors predict that there are more students in Velp pursuing "Care and Wellbeing" education area may influence the healthy food choices with slowly introduction of healthy food choices? If so, please provide information in the Discussion section.
- Page 7, line 294 -- key other outcome measures.
Do authors mean "other key outcome measures"?
- Page 9, table 3 and Table 4 -- Unstandardized constant
Can authors also include standardized constant in the table?
- Page 11, Table 5 Outcome measures, and Table 6 Descriptive
Both Tables need table headings that will describe the content of the tables, therefore, the tables can stand alone.
- Page 12 -- % Healthier choice products sold of total sales (total number sold)i
Does this superscript "i" refer to the footnote (i.e. Separate sweet and savoury bread toppings.....)? If so, please add "i" to the footnote.
Author Response
We want to thank reviewer for taking the time to review our paper. We appreciate the positive evaluation of our paper. The paper has been revised in response to the reviewers’ comments and we feel that the paper has benefitted greatly from the suggestions. We made several changes to improve the presentation of results. For example, we changed the sequence of tables to make the flow more logical and improved the lay-out of figure 1 and 2. Please find our specific response below.
Page 2, line 60-- ... recommend levels of healthy relative to unhealthy foods on display from.. Can authors expand to include the definitions for what determining the "healthy" and "unhealthy" foods?
Response : We changed the sentence in this paragraph (line 67): ‘In this study, we build on the guidelines of the Dutch Healthy Canteen Programme (developed by Netherlands Nutrition Centre) that define and recommend levels of healthy relative to unhealthy foods on display from 60% to 80%, based on Dutch national nutritional guidelines [12].’ More details about this programme and the determination of what is healthy and less healthy can be read in section 2.2. (line 135-148).
Page 3, line 103 -- This study was conducted from September 2016 until July 2017 at two vocational school canteens. This noted study period is inconsistent with the period noted in Table 1 (September till June - 10 months). Do authors consider "seasonality" issue for fresh produce availability? Was this an issue at that geographic location?
Response : Based on a comment of reviewer 1, we corrected the inconsistency in period indication in table 1 (see response 5).
We do not consider ‘seasonality’ issues as there was not much fresh produce available (i.e. some fruit and snack vegetables) and this is generally available throughout the year.
Page 3, line 107 -- Instead, students bring their own packed lunch or purchase meals, snacks and drinks at school or at food outlets nearby school. Would authors be able to comment on the price differences between the food purchased from school vs. food purchased from food outlets nearby schools? Would this influence on food purchasing behavior?
Response : We added the following sentence to line 118: ‘Typically, prices at schools commercial school canteen catering companies are a bit higher than in supermarkets or fast food chains.’ We also commented on this price issue in the discussion: ‘This is typical for many canteens at secondary and vocational schools in the Netherlands, and partly driven by high price perceptions of the foods sold [31].’
Page 3, line 126 -- ... when they are healthy or reasonably healthy. Can authors provide some examples for those food category?
Response : We now provide examples at line 139: ‘Examples of such products are brown or whole wheat bread, dairy drinks without added sugars and salads’.
Page 4, line 146 -- In the first baseline phase (September until November 2017), the visible...Do authors mean "2016"?
Response : No, this is also 2017 and we changed this in the text.
Page 4, line 150 and page 5 line 169 -- ... when their stock is too low../ ... on display was too low, ....Can authors provide the definition of "too low"? Was there any issue on supplying healthy or unhealthy food at school?
Response 19: There was no issue on supplying healthy or unhealthy food at school. What we mean is that visible share of products was manipulated in that relatively more healthy products are meant to be on display for customers than unhealthy products. However, when students purchase a lot of particular food, canteen employees can refill these items. The word ‘out of stock’ seems less appropriate, as the canteen has enough in their own storage. We therefore changed it into ‘when availability is too low’ or ‘products that got low in availability’.
Page 6, line 272 -- Differences between product groups. Can this paragraph be moved to the section "Canteen choice characteristics" (staring line 244)?
Response : We moved this paragraph up.
Page 7, table 2 -- Type of education of "Care and Wellbeing" followed for school Ede and Velp are 148 and 204, respectively. Do authors predict that there are more students in Velp pursuing "Care and Wellbeing" education area may influence the healthy food choices with slowly introduction of healthy food choices? If so, please provide information in the Discussion section.
Response : It may indeed have impacted the results. We included a sentence in the part of the discussion where we mention an important limitation (line 436): ‘Furthermore, an important limitation is that there were no control canteens in which there were no guidelines on the type and size of the offered assortment. Moreover, different types of students in terms of type of education were present at the two school locations. This study shows what happens when basic entry levels of 60% healthier assortment are exceeded’.
Page 7, line 294 -- key other outcome measures. Do authors mean "other key outcome measures"?
Response : Yes indeed and we changed this mistake.
Page 9, table 3 and Table 4 -- Unstandardized constant. Can authors also include standardized constant in the table?
Response : A standardized constant in a regression analysis is particularly useful when using a variety of unstandardized measurement scales. In that case, you would get an intercept of zero. In our logistic regression we have a standardized dependent choice variable (0 for healthier choice and 1 for unhealthy choice), and a set of dummies. Therefore, we feel that it is only valuable to report the unstandardized values and the intercept of the logistic regression equation to help interpretation of the results. Particularly this intercept is relevant as it shows what happens when all predictors are zero. These are included in table 5 and 6.
Page 11, Table 5 Outcome measures, and Table 6 Descriptive. Both Tables need table headings that will describe the content of the tables, therefore, the tables can stand alone.
Response : We changed the table heading of table 4 (used to be table 6) into: ‘Table 4. Descriptives of experimental design, sales, revenues and number of surveyed participants in each of the four periods in the two canteens’. We also changed the table heading of table 3 (used to be table 5) into: ‘Table 3. Outcome measures of cash register and survey data in each of the four periods in two canteens’ .
Page 12 -- % Healthier choice products sold of total sales (total number sold)i Does this superscript "i" refer to the footnote (i.e. Separate sweet and savoury bread toppings.....)? If so, please add "i" to the footnote.
Response : The subscript 1 refers to the footnote 1.
Round 2
Reviewer 1 Report
I accept the authors' extensive explanations. The authors have included almost all suggested changes. They know the limitations resulting from the conducted treatments, which may be a premise for conducting other surveys/analyzes.
Please keep in mind that there is still a minor mistake which was indicated in my 5th comment (a phrase repetition) and it is not addressed by the authors (probably they have misunderstood the meaning of my comment). What I meant is that in lines 217-218 (now in the revised manuscript) says: answered on a five-point Likert-type response format ranging from ‘totally disagree’ to ‘totally agree’. Then in lines 221-222 the above phrase is repeated and says: All items were answered on a five-point Likert-type response format ranging from ‘totally disagree’ to ‘totally agree’. Probably you should keep one of them in this paragraph.
Author Response
We thank the reviewer for the careful inspection of the revised document. We now removed one of the sentences. Indeed, this was a misunderstanding.